# Differential Paralog-Specific Expression of Multiple Small Subunit Proteins Cause Variations in Rpl42/eL42 Incorporation in Ribosome in Fission Yeast

**DOI:** 10.3390/cells11152381

**Published:** 2022-08-02

**Authors:** Wenzhu Li, Jing Zhang, Wenpeng Cheng, Yuze Li, Jinwen Feng, Jun Qin, Xiangwei He

**Affiliations:** 1MOE Key Laboratory of Biosystems Homeostasis & Protection and Innovation Center for Cell Signaling Network, Life Sciences Institute, Zhejiang University, Hangzhou 310058, China; wenzhuli@zju.edu.cn (W.L.); 21707055@zju.edu.cn (J.Z.); 11518033@zju.edu.cn (W.C.); lytze@zju.edu.cn (Y.L.); 2Human Phenome Institute, Fudan University, Shanghai 200433, China; jinwenf@fudan.edu.cn; 3State Key Laboratory of Proteomics, Beijing Proteome Research Center, National Center for Protein Sciences (The PHOENIX Center, Beijing), Beijing Institute of Lifeomics, Beijing 102206, China; jqin1965@126.com

**Keywords:** ribosome paralogs specificity, ribosome heterogeneity, ribosome concentration model

## Abstract

Ribosomes within a cell are commonly viewed as biochemically homogenous RNA–protein super-complexes performing identical functions of protein synthesis. However, recent evidence suggests that ribosomes may be a more dynamic macromolecular complex with specialized roles. Here, we present extensive genetic and molecular evidence in the fission yeast *S. pombe* that the paralogous genes for many ribosomal proteins (RPs) are functionally different, despite that they encode the same ribosomal component, often with only subtle differences in the sequences. Focusing on the *rps8* paralog gene deletions *rps801d* and *rps802d*, we showed that the mutant cells differ in the level of Rpl42p in actively translating ribosomes and that their phenotypic differences reside in the Rpl42p level variation instead of the subtle protein sequence difference between Rps801p and Rps802p. Additional 40S ribosomal protein paralog pairs also exhibit similar phenotypic differences via differential Rpl42p levels in actively translating ribosomes. Together, our work identifies variations in the Rpl42p level as a potential form of ribosome heterogeneity in biochemical compositions and suggests a possible connection between large and small subunits during ribosome biogenesis that may cause such heterogeneity. Additionally, it illustrates the complexity of the underlying mechanisms for the genetic specificity of ribosome paralogs.

## 1. Introduction

The ribosome is a mega-ribonucleoprotein complex that translates mRNAs into proteins and thus plays a central role in all organisms. The biochemical composition of the ribosome is well-defined and highly conserved. Ribosomes are seen as homogeneous molecular machines—thousands of mature ribosomes within a cell are conventionally regarded as identical.

However, this view is now challenged in light of emerging evidence and the reexamination of previous observations supporting heterogeneity in ribosome composition, including variant rRNA alleles, the incorporation of RP paralogs, and alterations in RP stoichiometry, as well as post-transcriptional rRNA and post-translational RP modifications (PTMs) (reviewed in [1]). The implications of its biochemical heterogeneity are the distinct ribosome types or “specialized ribosomes” that may influence gene expression through the selective translation of specific mRNAs or via modulating the translational efficiency of certain mRNAs, rendering ribosomes capable of serving as a hub for signal integration at the post-transcriptional level [2].

Tangible biochemical evidence supporting the ribosome heterogeneity hypothesis can be traced back to early studies of the ribosome composition [3]. In the 1970s, ribosomes purified from bacteria grown under specific conditions were shown to lack certain RPs but retained translational functions [4]. The comparison of ribosomes purified from rat skeletal muscle and liver using 2D gel electrophoresis revealed differences in the ribosomal composition between them [5]. In the amoeba *Dictyostelium discoideum*, ribosomes purified from spores and vegetative cells differ both in protein composition and posttranslational modifications [6]. More recently, using mass spectrometry, different stoichiometry of core RPs were detected in yeast cells grown under different growth conditions [7]. Exposing yeast to high salt or high pH causes a fraction of ribosomes to become Rps26-deficient [8]. In mouse embryonic stem cells (ESCs), four RPs: Rpl10A/uL1, Rpl38, Rps7, and Rps25, were present at the sub-stoichiometric levels determined by selected-reaction monitoring-based mass spectroscopy [9].

Recent advances have demonstrated the functional specializations of heterogeneous ribosomes. For example, distinct mRNA subsets were found to selectively associate with ribosomes enriched for Rpl10A/uL1 [9]. In addition, Rps26/eS26-containing ribosomes selectively translate certain transcripts by recognizing the Kozak sequence. The accumulation of Rps26/eS26-deficient ribosomes in yeast was shown to be part of the response to multiple stresses [8].

Ribosomes can also be specialized through the incorporation of different paralogs. In *E.coli*, as cells undergoing transition from the exponential to stationary growth phase, ribosomal core proteins bL31A and bL36A are replaced by paralogous bL31B and bL36B, respectively [10]. In the budding yeast *S. cerevisiae*, for two-thirds of the 78 RPs, each is encoded by a pair of independent paralog genes. Despite their high sequence similarity, deletions in RP paralogs often exhibit different phenotypes in a divergent array of functional outputs [11,12,13], demonstrating that RP paralog genes may be functionally different. One plausible interpretation is the “ribosome code” hypothesis, which postulates that heterogeneity in the ribosome composition by incorporating divergent RP paralogs (i.e., specialized ribosomes) may cause different functional outputs, perhaps by selectively translating specific subsets of mRNAs [11,14]. According to this model, the same transcriptome can be translated differently depending on variations in the “decoding” machinery—the specialized ribosomes.

The “ribosome code” hypothesis, however, does not exclude other models for the paralog-specific phenotypes. For example, the in-depth analysis of specific RP paralogs has shown that Rpl32/eL32, Rpl2a/uL2, Rps14b/uS11, and Rps28b/eS28 may function outside the context of ribosomes (called extra-ribosomal functions), and therefore, the paralog-specific phenotypes of these RPs do not necessarily suggest ribosomal heterogeneity [15,16]. Importantly, an alternative, “ribosome concentration” model proposes that a global reduction in ribosomes could impact mRNAs differently, significantly attenuating the translation of the poorly initiated mRNAs compared with the efficiently initiated ones [17,18]. Such a scenario could also explain the paralog-specific deletion phenotypes, due to the fact that most genetic manipulations of ribosomal compositions also alter the ribosome concentration simultaneously [19]. Furthermore, changing the expression of one paralog of a RP often affects the expression of the other [18].

Overall, whether paralog-specific phenotypes are due to RP paralogs with disparate functions (ribosome heterogeneity model) or due to differences in the level of ribosomes (ribosome concentration model) or a combination of both has not been tested systematically.

In this study, we conducted a genetic survey in a large collection of ribosome paralog gene deletion strains of the fission yeast *S. pombe*. We observed extensive paralog-specific phenotypes in multiple functional outputs, including centromeric chromatin epigenetic stability (CEN-PEV), cycloheximide sensitivity, and responses to stress conditions such as amino acid starvation. Focusing on deletions of the Rps8/eS8 paralog pair (*rps801d* and *rps802d*) that exhibit the highest contrast in these phenotypes, we showed that the mutant cells differ in the level of Rpl42p/eL42 in actively translating ribosomes, and this difference in Rpl42p/eL42, instead of the difference in protein sequences of the Rps8 paralogs, underlies the phenotypic differences in *rps801d* and *rps802d*. Furthermore, we found three other pairs of *rps* paralog genes, the deletions of which also cause different levels of Rpl42p/eL42 incorporation into actively translating ribosomes. Together, our work suggests Rpl42p/eL42 may be a major source of ribosome heterogeneity and illustrates the complexity of the underlying mechanisms for ribosome paralog phenotype specificity.

## 2. Materials and Methods

### 2.1. Yeast Strains, Genomic Manipulations, and the Culture Conditions

The *S. pombe* strains used in this study are listed in Appendix A. Yeast cells are grown on YE + 5S medium (with 5 supplements added, including histidine, uracil, lysine, leucine, and adenine) or YE + 4S medium (with histidine, uracil, lysine, and leucine, except adenine is provided by the yeast extract). Solid malt extract (ME) medium is used for mating and sporulation. Yeast strains are constructed by a tetra analysis. Gene deletions were performed by homologous recombination using the standard PCR-based amplification of deletion cassettes. Standard LiOAc-based protocols were used for transformations of plasmids and PCR products into yeast.

### 2.2. Construction of Homogenic and Chimeric Rps8/eS8 Strains

The ORF of each paralog, extending from the start codon to the stop codon, was amplified and cloned as pFA6a-Rps801-hph and pFA6a-Rps802-hph. Then, the two different amino acids (130aa and 133aa) in one paralog was exchanged for the same residues of the other paralog by site-directed mutagenesis. Successful mutagenesis was confirmed by sequencing. The resulting chimeric alleles (*rps801mm (S130A N133T)* and *rps802mm (A130S T133N)*) consisted of the protein sequence of one paralog, and the 5′UTR, 3′UTR, promoter, and terminator were denoted as the “locus”, of the other. Mutants harboring a chimeric allele were constructed by precisely replacing the ORF of Rps801 or Rps802 by homologous recombination using standard PCR-based amplification. The resulting “homogenic” strains harbor the same protein sequence as both the native allele and a chimeric *rps801mm (S130A N133T)* and *rps802mm (A130S T133N)* allele. Subsequently, the Rps801 or Rps802 ORF was deleted at its native locus in the homogenic strains to yield “chimeric” strains.

### 2.3. Spotting Assay

For the spotting assay, serial dilutions (10-fold) of the different strains in the growth medium were performed before by plating drops onto YE5S and YE5S with different concentrations of drugs (cycloheximide, Solarbio, Beijing, China) or NH_4_Cl added at 29 °C. Plates were photo-documented after 3d, respectively.

### 2.4. RNA Isolation and Quantitative Reverse Transcript Real-Time PCR (qRT-PCR)

RNA was extracted from yeast strains using the Hipure yeast RNA kit (Magen, Guangzhou, China) with a DNase I treatment following the manufacturer’s protocols. cDNA synthesis was performed using oligo (dT) and the Revert Aid first strand cDNA synthesis kit (Thermo Fisher Scientific, Waltham, MA, USA).

All real-time PCR were done with Bio-Rad CFX96 Touch. All samples were run in triplicate to ensure the accuracy of the data, and their average was calculated. PCR of 45 cycles was done using the SYBR Green qPCR kit (Bio-Rad172-5120). Primers were used at 0.3 uM for each experiment. The primer sequences for qPCR are available upon request.

### 2.5. Relative mRNA Expression Level of Rps8/eS8 Paralogs by cDNA Sequencing

The specific PCR products (~200 bp) containing the differential DNA sequence of Rps8/eS8 paralogs were amplified from wild-type cDNA under normal conditions. Multiplexed libraries were prepared at the same time using the library preparation kit and the barcode adapters (NEB). All the libraries were sequenced on Illumina, and approximately 0.1 million aligned reads per sample were taken. Count the number of sequencing reads that could align accurately with the DNA sequence of Rps801 or Rps802, respectively. Then, the mRNA level of the Rps8/eS8 paralogs in wild-type cells were shown by the ration of the Rps8/eS8 paralogs reads number among the total reads number.

### 2.6. Polysome Fractionation and Protein Extraction

Raise cells to OD_600_ = 1.2~1.5. Add cycloheximide to the yeast culture at a final 100 uM for 30 min before harvest cells, and resuspend cells with lysis buffer (50 mM Tris-Acetate, 50 mM NH_4_Cl, and 12 mM MgCl added to 1 mM DTT and 100 ug/mL cycloheximide, 200 U/mL RNase inhibitor (Thermo Fisher Scientific, Waltham, MA, USA), and 1× combined protease and phosphatase inhibitor (Thermo Fisher Scientific, Waltham, MA, USA)). The cell suspension was frozen in liquid nitrogen to form the small droplets by using the pipette. The cell droplets were lysed in a Tissuelyser-48 machine with 30 HZ 180 s for 6 times. The lysate was clarified by sequential centrifugation for 10 min at 4000× *g* rpm and 20 min at 12000× *g* rpm at 4 °C to remove the nuclei and mitochondria. Samples containing 100 OD260 units in 200 uL were loaded onto 11 mL 10–50% (*w*/*v*) sucrose gradients (100 mM Tris-Acetate, 100 mM NH_4_Cl, and 12 mM MgCl added to 1mM DTT and 100 ug/mL cycloheximide). Samples were centrifuged in a Beckman SW41Ti rotor at 39,000 rpm for 2.5 h at 4 °C, and 0.75-mL fractions were collected from the top of the gradients with a Brandel fractionator system. Traces were recorded on a UA6 detector at a sensitivity setting of 0.2. The proteins in each fraction were TCA precipitated and resuspended in 30 uL of a sample buffer (1× Laemmli buffer with bromophenol blue and DTT) kit. The A_254_ gradient profiles were digitized using a DATAQ DI-148U data recording module that converts and exports analog absorbance readings to analysis software.

### 2.7. Western Blotting

Protein samples were loaded onto 15% SDS-PAGE gel. After running, proteins were transferred, and the PVDF membranes were blocked in 5% nonfat dry milk in TBS for 1 h and incubated overnight at 4 °C with the following dilutions of primary antibodies: Rps8/eS8 (1:1000), Rpl42/eL42 (1:2000), and Rpl27/eL27 (1:500) (Lab stock), then incubated with HRP-conjugated donkey anti-rabbit (1:4000) secondary antibodies (Thermo Fisher Scientific, Waltham, MA, USA), and antigen was detected using super-signal west pico plus chemiluminescent substrate (Thermo Fisher Scientific, Waltham, MA, USA).

### 2.8. Mass Spectrometry and Data Analysis

Cells of wild-type, *rps801* deletion, and *rps802* deletion strains were grown in YE5S liquid medium. Yeast cells were lysed, fractionated by SDS-PAGE, and in-gel-digested with trypsin, as described previously. Each fraction was subsequently analyzed by online liquid chromatography-tandem mass spectrometry (LC-MS/MS). Three biological repeats were prepared for each strain. For each strain, six divides were taken from each prepared samples for parallel LC-MS/MS runs. All LC-MS/MS experiments were performed on an LTQ-Orbitrap mass spectrometry connected to an Agilent 1200 nanoflow HPLC system by means of a nanoelectrospray source. MS full scans were acquired in the Orbitrap analyzer using internal lock mass recalibration in real time, whereas tandem mass spectra were simultaneously recorded in the linear ion trap. Peptides were identified from MS/MS spectra by searching them against the yeast ORF database using the Mascot search algorithm.

With the help of the Firmiana platform Qin lab, source files were processed into mapped protein quantification values such as area, iBAQ, and a fraction of the total (in this conduction, the iBAQ value was used for processing), where the data from every six divides of one biological repeat are processed into one table. The iBAQ algorithm sums all identified peptide intensities and normalizes them against detectable tryptic peptides between 6 and 30 amino acids in length [20]. The quantification data were normalized using median values of the whole proteome. The multiple Student’s *t*-test was preformed between groups.

### 2.9. Microscopic Observation

Logarithmic growth cells carrying Rpl3201-GFP were imaged using DeltaVision Core and Personal DV (Applied Precision, Rača, Slovakia). Optical sectioning at 0.3 mm/section × 20 sections was performed. Z-stack images were deconvolved, projected using the Maximum protocol to generate a 2D image.

### 2.10. Statistical Analysis

All experiments were performed in 3 biological replicates, unless stated otherwise. Data are presented as the mean ± SD. Statistical analysis was made for multiple comparisons using the analysis of variance and Student’s *t*-test or Fisher’s exact test. A *p*-value <0.05 was considered to be statistically significant.

## 3. Results

### 3.1. Paralog-Specific Phenotypes Are Observed in Multiple Pairs of Ribosome Paralog Deletions in Fission Yeast

We initially aimed at elucidating the genetic pathways that influence the epigenetic stability of the centromeric chromatin organization in fission yeast [21]. To this end, we performed a genetic screen in an annotated gene deletion haploid strain collection, using an established centromere position-effect variegation (CEN-PEV) system as the readout [22]. In this system, the variegated expression of the ade6 reporter gene inserted in Centromere 2 (cnt2::ade6) is visualized by the coloration of individual colonies grown with a low supply of adenine. The white or red color of the colony corresponds to the ON or OFF state of ade6 expression, respectively. We previously have shown that CENP-A/Cnp1 nucleosome occupancy on ade6 directly correlates with its silencing and that the CEN-PEV effect reflects the epigenetic stability of Cnp1 nucleosome positioning within the centromere throughout mitotic cell generations [23]. We reasoned that any mutation altering CEN-PEV implies that the mutated gene is functionally involved in regulating centromere epigenetic stability directly or indirectly. Based on the altered coloration patterns of the colonies, almost half of the tested RP deletions (27/54) were found to change the colony coloration pattern.

In the pombe genome, 140 genes encode 78 ribosomal proteins (RPs) (http://ribosome.med.miyazaki-u.ac.jp, accessed on 11 December 2007), with 56 RPs encoded by two or more paralogous genes. Among 21 paralog pairs tested in this study, 10 pairs exhibited a paralog-specific phenotype in CEN-PEV (Appendix A), suggesting that these paralogous genes differently impact centromere epigenetic stability. Such broad paralog specificity in RPs is in agreement with the findings in the budding yeast S. cerevisiae, in which various phenotype-based RP paralog specificities were identified and have led to the hypothesis of the “ribosome code” [11].

To test whether the functional differences between pombe RP paralogs are in the context of ribosomes or not, we further characterized the phenotypes of the paralog deletions in cellular processes that are directly linked to ribosomal functions.

It is known that, in fission yeast, high-quality nitrogen sources such as NH_4_^+^ inhibit the uptake of poor nitrogen sources such as amino acids [24]. For auxotrophic strains leu1-32 and ura4D-18, which cannot synthesize leucine and uracil, respectively, excessive NH_4_^+^ effectively causes leucine or uracil starvation. To compare their tolerance to starvation, we assessed the growth rates of the RP paralog deletions (under the proper auxotrophic genetic background) on YE media supplied with excess NH_4_Cl. In comparison to wild-type cells, the mutants were categorized into three types by their responses to excess NH_4_Cl: wild type-like, more resistant, or more sensitive to NH_4_Cl than WT. Among 26 pairs tested in total, 13 pairs with sister paralog deletions were sorted into different categories (Appendix A), suggesting that, for these 13 RPs, the paralogs function differently in ribosomes so that their deletions exhibit different responses to amino acid starvation.

We also examined the growth of the RP paralog deletion mutants on YE media supplied with sublethal concentrations of cycloheximide, a potent protein synthesis inhibitor that blocks translational elongation by interfering with tRNA translocation on the ribosome. We found that, in 10 pairs, the paralog deletion mutants exhibited different levels of sensitivity (Appendix A, 16 pairs tested in total).

Noticeably, Rps8/eS8, Rps11/uS17, Rps23/uS12, Rps28/eS28, Rpl28/uL15, and Rpl43/eL43 exhibited significant paralog specificity in the phenotypes by all three surveys above. Thus, consistent with previous studies on the budding yeast, our genetic study in fission yeast revealed that RP paralog specificity broadly exists in multiple biological processes in this organism, probably in the context of ribosome functions.

Among the tested RPs, the rps8 paralog deletions overall exhibited the highest contrast in phenotypical difference (Figure 1A and Appendix A). In the CEN-PEV assay, rps801d colonies showed a significant reduction in the frequency of color switching, while rps802d was wild type-like. Upon amino acid starvation, rps801d was moderately sensitive, while rps802d was resistant to amino acid starvation in comparison to the wild type. In addition, rps801d exhibited a higher sensitivity to cycloheximide than the wild type or rps802d; the latter two were indistinguishable under the tested conditions. We thus focused on rps8 paralogs to further investigate the molecular mechanisms underlying their phenotypic divergence.

### 3.2. Rps8 Deletions Impact on the Level of Large Subunit Protein Rpl42p/eL42 in Actively Translating Ribosome in a Paralog-Specific Manner

We hypothesized that the RP paralog deletions above, via differentially affecting the protein synthesis machinery, cause specific changes in the proteome, which eventually lead to the distinct phenotypes. We wished to identify the target proteins whose abundances were altered differentially due to rps8 paralog deletions. To do so, we quantified the relative abundance of individual proteins in whole-cell protein extracts, comparing between wild type and rps801d or rps802d, respectively, using quantitative mass spectrometry (LC-MS/MS; the details are described in Section 2).

Proteins whose abundances were significantly altered (≥two-fold) in rps8 paralog deletion strains were shown (Appendix A). Interestingly, the ribosomal proteins that were most frequently detected by mass spectrometry consistently exhibited significant alterations in abundance. Specifically, the abundances of all the small subunit proteins (RPSs) were consistently decreased in both rps8 paralog deletions compared to the wild type, whereas that of most large subunit proteins (RPLs) had no significant change (Figure 1B). The sole exception of this trend was the large subunit component Rpl42p/eL42, whose abundance was increased significantly in rps801d but decreased in rps802d compared to the wild type.

In order to determine whether the differential Rpl42p/eL42 protein levels in rps8 paralog deletions reflected the differences in the ribosome compositions, cell lysates were subjected to a polysome profiling analysis by sucrose gradient ultracentrifugation, separating free RPs from those incorporated into ribosome subunits (40S and 60S) or 80S ribosomes and polysomes. Polysome profiling detected a diminished level of 40S small subunits but a prominent accumulation of 60S large subunits in both rps801d and rps802d compared to the wild type (Figure 1C and Appendix A). This is consistent with a recent report of the accumulation of RPLs in the budding yeast rps mutants, although the accumulation of 60S there does not seem as drastic [25,26]. Firstly, the Rpl42p/eL42 levels in each fraction separated by polysome profiling were determined by Western blotting using an antibody specifically against Rpl42p/eL42. The result showed that the Rpl42p/eL42 protein is present and accumulated in the 60S fraction, as well as 80S and polysome fractions in both rps8 paralog deletions (Appendix A). Next, the Rpl42p/eL42 levels were measured in the whole-cell lysate or actively translating ribosomes. The result showed that the Rpl42p/eL42 level in the polysome fractions was increased in rps801d compared to that in the wild type but decreased in rps802d (Figure 1D), suggesting the Rpl42p/eL42 level is varied in actively translating ribosomes between rps8 paralog deletions. Intriguingly, no significant difference in the total Rpl42p/eL42 level could be detected consistently in the whole-cell extracts of rps8 paralog deletions by Western blotting, which might be due to compensatory variations in free Rpl42p/eL42 in the cytoplasm and/or technical differences between mass spectrometry (Figure 1B) and Western blotting.

Together, these results demonstrate that, while deletions of Rps8/eS8 paralogs indiscriminately affect the overall ribosome assembly at comparable levels, they impose opposite effects on the level of Rpl42/eL42 incorporation in intact ribosomes.

### 3.3. Differential Rpl42/eL42 Levels Contribute to Paralog-Specific Sensitivity to Amino Acid Starvation in rps8 Deletion Mutants

To test whether differential impacts on the Rpl42/eL42 level in ribosomes by rps8 paralog deletions may account for their phenotypical differences, we took a genetic approach to investigate the possible functional connection between Rpl42/eL42 and Rps8/eS8 paralogs.

We first constructed a heterozygous rpl42 deletion (rpl42^+/−^) diploid strain with the standard yeast molecular genetic methods. We induced meiosis in rpl42^+/−^ and performed a tetrad dissection for the meiosis progeny. The result showed that rpl42 gene deletion is lethal in haploids, demonstrating that rpl42 is an essential gene (Appendix A), in contrast to the result derived from systematic genome-wide gene deletion characterization [27].

We speculated that different levels of Rpl42/eL42 in ribosomes may be the primary molecular alteration that eventually causes phenotypical differences in Rps8/eS8 paralog deletion mutants. To this end, we tested whether altering rpl42 expression was sufficient to mimic the phenotypes of rps8 paralog deletions or not. We first measured the Rpl42/eL42 mRNA level in rpl42^+/−^ diploid by qRT-PCR and found that it was decreased to about 70% of the wild-type diploid level (Figure 2A). Consistently, the Rpl42/eL42 protein level measured in rpl42^+/−^ cell extracts by Western blotting was also decreased to a comparable degree (Figure 2B). Growing in YE supplied with excess NH_4_Cl, the rpl42^+/−^ heterozygous diploid strain was resistant to this stress, displaying enhanced growth compared to the wild-type diploid strain (Figure 2C). Conversely, the ectopic expression of rpl42 by inserting an extra copy of rpl42 ORF with its native promoter at the ura4 site (ura4::rpl42) strain reversed the expression level of rpl42 in the heterozygous rpl42^+/−^ diploid to the wild-type level. In addition, cell growth was reversed to the level of wild-type diploid cells with excess NH_4_Cl.

In order to test the genetic interaction between rps8 paralog deletions and rpl42, we deleted one copy of the rpl42 gene in rps801^−/−^ homozygous diploid and rps802^−/−^ homozygous diploid. Consistent with the rps8 paralog deletion haploid cells, the rps801^−/−^ diploid cells showed sensitivity to excess NH_4_Cl, and rps802^−/−^ diploid cells showed resistance to excess NH_4_Cl (Figure 2D). A single deletion of the rpl42 gene (rpl42^+/−^) reversed the growth of rps801^−/−^ diploid cells in YE supplied with excess NH_4_Cl. Ribosomal proteins, such as Rpl36A (homolog of Rpl42/eL42), cannot be overproduced in budding yeast, because the excess proteins are rapidly degraded [28]. Consistently, while an extra copy of rpl42(ura4::rpl42) in wild-type fission yeast haploid cells did cause increased rpl42 transcription (Appendix A), nonetheless, no increase was detected in the levels of the total Rpl42/eL42 protein or ribosome-integrated Rpl42/eL42 by Western blotting (Appendix A). This precluded the test on whether increased Rpl42/eL42 incorporation in ribosomes could mimic the phenotypes of rps801d.

Together, these results support the notion that changes in the Rpl42p/eL42 level in rps8 paralog deletions contribute mainly, if not solely, to the paralog-specific phenotypes in response to amino acid starvation.

To further explore the genetic interactions between rpl42 and rps8 paralogs, we constructed a rpl42 random mutation library by mutagenic PCR and replaced the endogenous rpl42 gene in rps801d or rps802d haploid cells, respectively, via the standard procedure of yeast transformation/homologous recombination. We then screened for rpl42 mutation alleles that were able to reverse the phenotypes or reduce the phenotypic contrast in amino acid starvation (excess NH_4_Cl in YE media) of rps801d or rps802d.

Among over 1000 transformants covered in each genetic screen, rpl42(G51S and, F58S) and rpl42(K100M and A103T) mutations were identified to specifically suppress the enhanced growth of rps802d cells under the condition of YE supplied with excess NH_4_Cl (Figure 2E). All of these rpl42 mutations by themselves exhibited a wild type-like growth rate on YE with excess NH_4_Cl. Together, these results showed that ribosome small subunit protein Rps8/eS8 paralogs have a strong genetic interaction with the ribosome large subunit protein Rpl42/eL42 in response to amino acid starvation.

### 3.4. Differential Transcription Level of Rps8/eS8 Paralogs but Not Protein Sequence Variation Results in the Divergent Phenotypes of Rps8/eS8 Paralogs

Rps801p and Rps802p vary in the protein sequence by only two residues—130S in Rps801p vs. 130A in Rps802p and 133N in Rps801p vs. 133T in Rps802p (Figure 3A and Appendix A). On the other hand, the mRNA expression levels of rps801 and rps802 differed (55% and 45%) in wild-type cells, measured by cDNA deep sequencing (see Section 2, Appendix A), presumably due to the difference in strength of their promoters. Thus, deletions of rps801 and rps802 would impose at least two types of changes. First, the protein identity of Rps8p within ribosomes is changed—a mixture of Rps801p and Rps802p for wild-type cells and pure Rps802p and Rps801p for rps801d and rps802d cells, respectively. Second, the total expression of rps8 may be reduced to different levels by paralog deletions.

To determine whether either of the two changes accounted for the phenotypical differences in the rps8 paralog deletion strains, we stepwise constructed a series of haploid strains in which Rps8p/eS8 was produced only in the Rps801p or Rps802p form (Figure 3B), with the gene dosage either maintained at the wild-type level in the “homogenic” rps8 strains or reduced in the “chimeric” rps8 strains (see below).

We replaced the two non-conserved residues (130aa and 133aa) in one paralog for the corresponding residues of the other by site-directed mutagenesis. The resulting mutation alleles rps801mm (S130A and N133T) and rps802mm (A130S and T133N) encode the protein sequences of Rps802p and Rps801p, respectively. These “homogenic” strains, rps801mm and rps802mm, each harbor two copies of the Rps802p- and Rps801p-coding sequences at both native loci.

On the other hand, for comparing paralogs at a single-copy dosage, we normalized the difference in the expression levels between the paralogs. To do so, the rps802 and rps801 ORFs were deleted at their native loci in the rps801mm and rps802mm homogenic strains to yield “chimeric” strains (rps801mm rps802d and rps802mm rps801d); each had only one type of Rps8p/eS8, Rps802p, and Rps801p protein expressed under the promoter of their paralogous sister (hence, chimeric), respectively (see Appendix A for the summary of the above strains).

Firstly, the Rps8/eS8 mRNA level of the above strains was measured by qRT-PCR (Figure 3C). The results showed that the Rps8/eS8 mRNA level was decreased in rps801d and rps802mm rps801d to about 40% of the wild type and, in rps802d and rps801mm rps802d, to about 60% of the wild type, respectively. It suggests that the deletion strains, as well as the chimeric strains, have a similar mRNA level to Rps8/eS8. In addition, the Rps8/eS8 mRNA level in the homogenic strains was comparable or slightly decreased (rps801mm in Figure 3C) compared to the wild-type strain.

We then measured the colony size as a proxy of the growth rate (Figure 3D). Both single rps8 paralog deletions and chimeric strains exhibited a decreased colony size compared with the wild type. No significant difference was detected between rps801d and rps801d rps802mm, rps802d, and rps802d rps801mm. Additionally, the colony sizes of rps802d and rps802d rps801mm were larger than those of rps801d and rps801d rps801mm. These data suggest that, in mutant cells harboring only one rps8 allele, the rps801 locus has a bigger impact than the rps802 locus on cell growth, regardless of the produced protein being Rps801p or Rps802p. For homogenic strains (rps801mm and rps802mm) with two copies of the same paralog, the colony size was indistinguishable from that of the wild type.

These findings indicate that the genetic loci (and thereby, presumably, the promoter strengths) of the rps8 paralogous genes, rather than the particular Rps8/eS8 protein paralogs, affect the cell growth rate.

We then measured the Rpl42/eL42 levels of the actively translating ribosome (polysome fractions from polysome profiling) in the Rps8/eS8 homogenic and chimeric strains by Western blotting (Figure 3E,F). The mutants harboring only one rps8 allele consistently had diminished levels of the 40S subunits, decreased levels of 80S and the polysomes, and a drastic accumulation of 60S subunits relative to the wild type. On the other hand, the polysome profiles of the homogenic rps801mm and rps802mm strains were similar to that of the wild-type strain.

Western blotting showed that the level of Rpl42p/eL42 was indistinguishable between rps801mm, rps802mm and the wild type. The Rpl42p/eL42 protein level was more abundant in rps802mm rps801d (expressing Rps801p under the promoter of rps802) than the wild type and even more than rps801d, which expressed Rps802p under the promoter of rps802. In addition, the Rpl42p/eL42 protein level was less abundant in rps801mm rps802d (expressing Rps802p under the promoter of rps801) than the wild type and was indistinguishable from rps802d, which expressed Rps801p under the promoter of rps801 (Figure 3F). Thus, a reduction in rps8 expression to variegated levels caused the different levels in the Rpl42p/eL42 protein incorporation in the ribosomes among these strains.

In order to explore whether the differential expression level of Rps8 in the above mutants causes differential rpl42 transcription, which ultimately leads to different levels in the Rpl42p/eL42 protein incorporation into ribosomes, we measured the Rpl42/eL42 mRNA level in the above strains by qRT-PCR (Appendix A). However, no significant change in the rpl42 transcription level was detected in these mutants in comparison to the wild type.

Lastly, we determined whether the varied Rpl42p/eL42 protein levels caused by differential Rps8/eS8 transcription levels would also affect the responses to amino acid starvation. We examined the growth of these strains on YE media supplied with excess NH_4_Cl (Figure 3G) and found that the growth of both rps801mm and rps802mm homogenic strains were comparable to that of the wild type. rps802mm rps801d displayed the highest sensitivity to amino acid starvation among the tested strains, since the Rpl42p/eL42 protein level was the most abundant in this strain by the previous Western blotting results (Figure 3F). In contrast, rps801mm rps802d was resistant to amino acid starvation, similar to rps802d. Thus, the differential Rps8/eS8 transcription level, but not the specific Rps8p/eS8 protein identity, contributed to the divergent phenotypes between rps8 paralog deletions and chimeric strains.

Altogether, our results demonstrated that the cause for divergent phenotypes of rps8 paralog deletions resides in the differential rps8 expression rather than the variations in the Rps8/eS8 paralog protein sequence.

### 3.5. Additional 40S Ribosomal Protein Paralog Gene Deletions also Cause Differential Levels of Rpl42/eL42

We wished to investigate whether the variations in Rpl42/eL42 level were broadly related to paralog-specific phenotypes in other RPs or not in addition to Rps8/eS8. Since the genetic interaction between Rpl42/eL42 and Rps8/eS8 was clearly reflected by the cell growth phenotype in the presence of excessive NH_4_Cl, we reasoned that this phenotype might serve as an indicator for other RPSs with paralog-specific effects on the Rpl42p/eL42 level.

In this vein, a collection of RP paralog deletions (Appendix A) was surveyed for differential growth rates at the presence of excess NH_4_Cl. Compared to the wild type, rps1102d, rps2301d, and rps2801d were sensitive to excess NH_4_Cl, whereas their corresponding paralog deletions (rps1101d, rps2302d, and rps2802d) were resistant (Figure 4).

Next, the cell lysates of these strains were subjected to a polysome profiling analysis by sucrose gradient ultracentrifugation. Similar to rps8 paralog deletions, the abundance of 40S small subunits was diminished, whereas the 60S large subunits accumulated drastically in the rps11, rps23, and rps28 paralog deletions compared to the wild type (Figure 4A). The actively translating ribosomes (the polysome fractions) of all six deletion strains, as well as wild type, were collected, and Rpl42p/eL42 levels were assessed by Western blotting. Higher levels of Rpl42p/eL42 were detected in rps1102d, rps23d, and rps2801d cells than in rps1101d, rps2302d, and rps2802d, respectively (Figure 4B).

Furthermore, by introducing the rpl42(K100M and A103T) mutation in the rps11, rps23, and rps28 paralog deletion strains, the phenotype of the enhanced growth of rps1101d, rps2302d, and rps2802d in the excess of NH_4_Cl was reversed, similar to that observed in rps802d. This genetic evidence suggests that the variegated Rpl42p/eL42 levels in the paralog deletions of these RPSs are responsible for the paralog-specific phenotypes, similar to that in rps8 (Figure 4C).

According to the crystal structure of the yeast 80S ribosome [29], Rps8/eS8, Rps11/uS17, Rps23/uS12, and Rps28/eS28 are all located at the interface of the 40S and 60S subunits (Figure 4D). In order to test whether these RP paralog deletions are epistatic to the rps8 paralog deletions, we tested the growth of double mutants rps801d rps2801d, and rps802d rps2802d under the condition of YE supplied with excess NH_4_Cl. The results showed that the double mutants did not exhibit a greater sensitivity or resistance to excess NH_4_Cl than single mutants (Appendix A), suggesting that rps801d, rps2801d, rps802d, and rps2802d function in the same genetic pathway in response to amino acid starvation.

Rps11/uS17, Rps23/uS12, and Rps28/eS28 paralogous gene pairs encode the RP paralogs with identical sequences. This strongly suggests that the paralogs are unlikely to carry distinct biochemical activities to account for the differential incorporation of Rpl42/eL42. Rather, the variegated Rpl42/eL42 protein levels observed in these rps paralog deletions may be caused by differential RPS expression levels. Consistent with this, the single deletion of paralogous genes impacts the total expression level of the RP differently. By measuring the mRNA levels of each paralog upon deletions of its paralog partner in comparison to the wild-type cells (Appendix A), we found that the Rps11/uS17 mRNA level in rps1101d was decreased to about 90% of the wild type and, in rps1102d, to about 70% of the wild type, respectively. The Rps23/uS12 mRNA level in rps23d was decreased to about 50% of the wild type but increased in rps2302d to about 120% of the wild type, suggesting that the deletion of rps2302 may alter the expression of rps23.

Together, these results demonstrate that a variation in the Rpl42p/eL42 level is not restricted to rps8 paralog deletions but, rather, broadly underscores paralog-specific phenotypes of multiple RPS paralog deletions.

### 3.6. 60S Subunits Accumulation Is Independent to Varied Rpl42/eL42 Level in rps8 Paralogs

60S subunit accumulation was observed in rps deletion mutants, including rps8, rps11, rps23, and rps28 paralog deletion mutants (Figure 1, Figure 4 and Appendix A). This accumulation might result either from a block in the maturation of 60S subunits or from the cellular tolerance of super-stoichiometric mature large subunits relative to small subunits. Ribosome assembly follows a discrete order in which RPs are integrated into precursors of large and small subunits at specific stages in conjunction with the precursors migrating from the nucleolus to the nucleoplasm and, eventually, to the cytoplasm in the cell. Rpl42p/eL42 is integrated into pre-60S at a late stage, immediately before or after pre-60Ss are exported to the cytoplasm [30]. We wished to test whether 60S accumulation in the cytoplasm might contribute to differential Rpl42/eL42 incorporation in rps8 paralog deletions or not, perhaps by providing the time window necessary for creating such a difference.

First, the process of ribosome assembly was examined microscopically. The Rpl3201-GFP fusion protein, which is integrated into pre-60S at an early step in the nucleus [30], was used as an indicator for ribosome subunit precursors. In wild-type cells during exponential growth, nuclear GFP signal density represents the intermediates in the process of 60S large subunit assembly and is approximately half of the total Rpl3201-GFP in the cell (Figure 5A). If 60S maturation were blocked in rps8 mutants, the accumulation of Rpl3201-GFP in the nucleus could be observed. The results showed that the subcellular distribution of Rpl3201-GFP was not accumulated in the nucleus in rps8 paralog deletions. The Rpl3201-GFP signals in the nucleus vs. whole cell were indistinguishable between rps8 paralog deletions and wild-type cells (Figure 5A). It suggests that the accumulated 60S subunits detected by polysome profiling in the mutants were mature (or late-stage) large subunits and were exported into the cytoplasm. This is consistent with the mass spectrometry result in which the protein level of nearly all RPLs (except for Rpl42p/eL42) uniformly increased (Figure 1B).

We then tested whether alleviating the 60S subunits accumulation in rps8 paralog deletions would suppress their differential Rpl42p incorporation phenotypes or not. Here, a Rpl31/eL31-GFP fusion construct was tested, because Rpl31/eL31 in fission yeast is encoded by a single gene and that GFP tagging partially compromises the functions of Rpl31 and, thus, alters the kinetics of 60S assembly (Figure 5B, the reduced 80S peak of Rpl31-GFP in comparison to the wild type). The polysome profiling analysis showed that Rpl31/eL31-GFP indeed alleviates 60S subunit accumulation in both the rps801d and rps802d strains (Figure 5B). However, rpl31-GFP did not affect the growth of rps801d or rps802d at the tested conditions (Figure 5C). These results suggest that the suppression of 60S subunits accumulation is independent of the differential growth rates of rps8 paralog deletions in YE media supplied with excess NH_4_Cl and, by extension, may be independent of the varied Rpl42/eL42 protein levels, the primary determinant to the differential growth rates in these strains (Figure 1, above).

## 4. Discussion

### 4.1. Identifying Varied Levels of Rpl42p/eL42 Incorporation in Actively Translating Ribosomes

We found that the Rpl42p/eL42 level in actively translating ribosomes was higher in *rps801d* cells than in wild-type cells but lower in *rps802d* cells than in wild-type cells. In addition, we discovered that *rps801d* cells with a higher Rpl42p/eL42 level exhibited a higher sensitivity to cycloheximide than the wild type or *rps802d*. This is consistent with cycloheximide directly binding with eL42 to block the binding of E site tRNA or P/E hybrid-state tRNA and therefore inhibiting ribosome translocation during translation elongation [31]. Together, these results imply that Rpl42p/eL42 in actively translating ribosomes purified from wild-type cells may be at the sub-stoichiometry level. We speculate that ribosomes in wild-type cells may exist in two forms—those with and those without Rpl42p/eL42, which are probably at a balanced ratio. Accurate measurements of the sub-stoichiometric levels of Rpl42p/eL42 in ribosomes need to be carried out in the future by more quantitative approaches, such as quantitative proteomic mass spectrometry or cryo-EM on intact ribosomes in pombe cells at various genetic settings or physiological conditions.

Ribosomes varied in Rps25/eS25, Rps26/eS26, and Rpl10A/uL1 functions differently, suggesting a diversified ribosome heterogeneity [32]. Interestingly, Rps25/eS25, Rps26/eS26, and Rpl10A/uL1 are all located at the vicinity of the mRNA exit channel (E site) [8,9,33]. The L44e protein in *Haloarcula marismortui*, the structural homolog of Rpl42/eL42, is also located near the E site in the large ribosomal subunit [34] and interacts with an RNA oligonucleotide that mimics the CCA end of deacylated tRNA bound to the E site [35]. Recent evidence suggests that Rpl42/eL42 directly contributes to peptidyl-transferring enzymatic catalysis activity [36]. It is conceivable that the Rpl42p/eL42 level variation in actively translated ribosomes may impact ribosome activity significantly.

We propose that, in multiple *rps* paralog deletions (including *rps8*, *rps11*, *rps23*, and *rps28*), the balance between the Rpl42/eL42-containing ribosomes and Rpl42/eL42-depleted ones is skewed, albeit in different ways. For example, in *rps801d* cells, the balance is skewed towards the form with Rpl42p/eL42 and, in *rps802d* cells, towards the form without. How such differential skewing is achieved is currently unclear. Furthermore, whether at any physiological conditions or with any external stimuli the RPS expression levels do change still need future investigation. Although we have shown that artificially altering the *rpl42* expression is sufficient to phenocopy *rps8* paralog deletions, no significant change in the *rpl42* transcription level was detected in these mutants in comparison to the wild type (Appendix A), implying that mechanisms other than *rpl42* transcriptional modulation (for example, the strength of Rpl42p/eL42 association with the ribosome or the protein stability of Rpl42p/eL42) may be at play in these mutants.

In budding yeast, no difference in the RP composition of ribosomes was found among wild-type and Rps28/eS28 paralog (Rps28A and Rps28B) deletions by quantitative mass spectrometry [25]. In budding yeast, the level of *Rps28B* mRNA is controlled by an autoregulatory feedback loop [16,37]: the Rps28B protein directly binds to a de-capping complex via Edc3p, which then binds to *Rps28B* mRNA, leading to its de-capping and degradation. This feedback regulatory mechanism seems species-specific, as a motif required for mRNA binding is present exclusively in Edc3 proteins from yeasts of the *Saccharomycetaceae* phylum [37]. This may explain the discrepancy of the Rpl42/eL42 level regulation in the fission and budding yeasts.

According to the crystal structure of the 80S ribosome, Rpl42/eL42 is located near the E site on the 60S large subunit. Interestingly, Rps8/eS8, Rps11/uS17, Rps23/uS12, and Rps28/eS28 are all located at the interface of the 40S and 60S subunits, although they do not have a direct interaction with Rpl42p/eL42 (Figure 4D). Perhaps these RPS proteins have a common effect on Rpl42/eL42 incorporation mediated via interactions between the small and large subunits.

Alternatively, we speculate that a mechanism may be at play in the process of ribosome biogenesis by which the pathways for large and small subunit assembly cross-talk, akin to a ribosome quality check [38]. In contrast to the current model that the assembly of large and small subunits is independent to each other, such a cross-talking mechanism would allow the level of small subunits to quantitatively influence the process of large subunit assembly; specifically, the incorporation of Rpl42p/eL42 at a late stage. This mechanism is probably not simply a prolonged halt at the free 60S stage, as the accumulated 60S subunits are mature large subunits (Figure 5A). On the other hand, we found that the Rpl42p/eL42 protein is also present in the accumulated 60S subunit in either *rps8* paralog deletion (Appendix A). In addition, inhibiting 60S accumulation does not affect the phenotypic contrast between the *rps8* paralog deletions (Figure 5B,C). Therefore, our results indicate that such a cross-talk mechanism may execute its function at the steps of 80S ribosome assembly.

### 4.2. Implications for RP Paralog Specificity

#### 4.2.1. Intertwining of the “Specialized Ribosome” Model and the Ribosome Paralogs Gene Dosage Effect

The question of whether the diverse phenotypic consequences of deleting one RP paralog vs. the other can be explained by differences in the RP gene dosage or may serve as evidence for “specialized” ribosomes due to the distinct subsets of RP isoforms [7,11,13,39]. It is often technically difficult to distinguish between these two models unequivocally.

By manipulating the RP expression dosage and mutating the protein sequence separately, our current study shows that paralog-specific phenotypes of *rps801* or *rps802* deletions can be explained by differences in the mRNA level of Rps8/eS8 when only one Rps8/eS8 locus is expressed, and the consequential, varied effects of differential reductions in Rps8p/eS8 on the Rpl42p/eL42 protein level in ribosomes. The two-residue variations in the ORFs of *rps801* and *rps802* are irrelevant to their functional differences. Thus, in this case, the genetic evidence appears to fit well with the gene dosage model while refuting the ribosome heterogeneity model.

On the other hand, we further showed that deletions of the *rps8* paralogs affect the Rpl42/eL42 level in actively translating ribosomes differentially, which ultimately accounts for their phenotypic differences. The fact that variations of the Rpl42/eL42 level in ribosomes are functionally significant and may lead to the ribosome heterogeneity model. Thus, our work serves as a good example that the two models, although conceptually distinct, are not mutually exclusive and may be mechanistically intertwined.

#### 4.2.2. Less Complexity than the Theoretical Maximum Postulated by the Ribosome Code or Ribo-Sphere Concepts

We show that variations in the Rpl42/eL42 level are not unique to *rps8* paralog deletions but are commonly caused by paralog gene deletions of multiple RPSs. This indicates that the ribosome heterogeneity is much less than suggested by the theoretical maximum variations predicted by the “ribosome code” hypothesis or the “ribo-sphere” model. Instead, Rpl42/eL42 may represent a small subset of RPs, which could serve as the “node” to integrate alterations in ribosome compositions due to paralog-specific changes in expression of many other RPs. Supporting this speculation, sub-stoichiometric ratios deviating from the stereotypical 1:1 ratio have been identified in multiple RPs based on quantitative mass spectrometry [7]. The fact that Rpl42/eL42 situates near a critical functional site (the E site) of the ribosome well justifies its role as an integration node. It is tempting to contemplate whether other RP subunits at ribosome key functional sites may also act as the integration nodes for RP paralog specificities.

## Figures and Tables

**Figure 1 cells-11-02381-f001:**
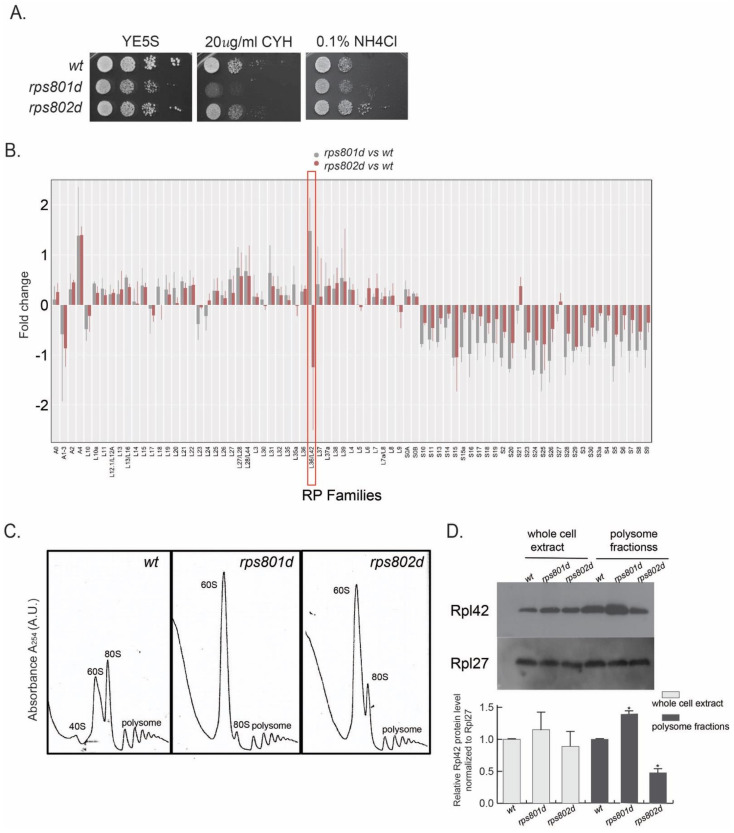
Rps8/eS8 paralog deletions exhibit different phenotypes upon various cellular process. (**A**) rps8 paralog deletions display paralog−specific phenotypes on the translation inhibitor drug (cycloheximide) and amino acid starvation (YE supplied with excess NH_4_Cl). (**B**) Quantification of ribosome proteins in rps8 paralog deletions, as well as wild types, by mass spectrometry. The details are described in Section 2. The fold change of Rpl42/eL42 (Rpl36A) exhibits an outstanding pattern. (**C**,**D**) Polysome analysis by sucrose gradient−based centrifugation among the wild−type, rps801d, and rps802d and quantification of the Rpl42/eL42 protein by Western blotting. Log−phase yeast grown on a rich medium supplemented with 100 uM cycloheximide prior to harvest in order to stabilize translating the ribosomes. The same amount of total cell extract was applied to sucrose gradient centrifugation. To localize the ribosomal subunits (40S and 60S) and intact ribosomes (80S and polysomes), fractionation was monitored at 254 nm. One−tenth of the total cell extract, as well as the protein extract from the polysome fractions, was immunoblotted. *p*−values were obtained by two−tailed Fisher’s exact test. * *p* < 0.05.

**Figure 2 cells-11-02381-f002:**
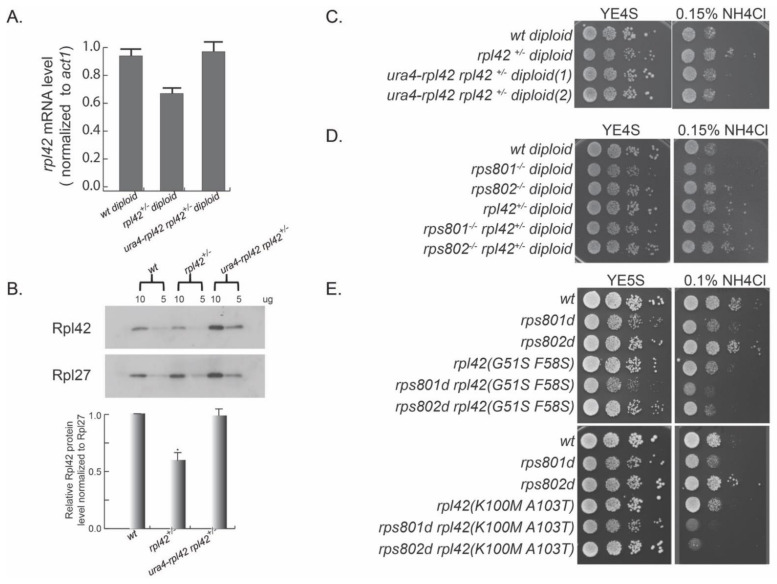
Differential Rpl42/eL42 protein level contributes to the paralog−specific phenotype in Rps8/eS8 paralog deletions. (**A**) Quantification of the relative Rpl42/eL42 mRNA levels in *rpl42^+/^*^−^ heterozygous diploid cells. (**B**) Quantification of the relative Rpl42/eL42 protein levels by Western blotting in diploid cell extracts. Ten or five micrograms of the total protein samples were loaded. *p*−values were obtained by the two−tailed Fisher’s exact test. * *p* < 0.05. (**C**) Reduced Rpl42/eL42 level in *rpl42^+/^*^−^ heterozygous diploid cells shows the enhanced growth in YE supplied with excess NH_4_Cl. (**D**,**E**) Rps8/eS8 paralog genes have strong genetic interactions with Rpl42/eL42.

**Figure 3 cells-11-02381-f003:**
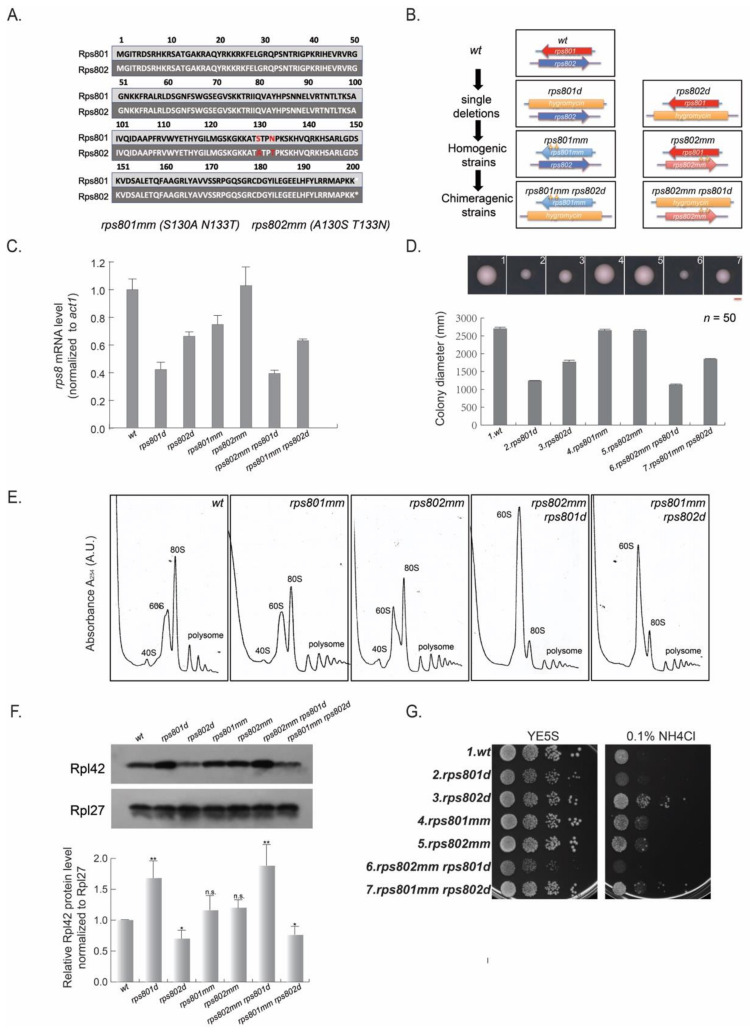
Differential transcription level of Rps8/eS8 paralogs results in the divergence phenotype of the Rps8/eS8 paralogs. (**A**) Comparison of the Rps8/eS8 paralog protein sequence. Rps801 and Rps802 differ in protein sequence by only two residues: 130aa and 133aa—the 130aa: S in Rps801p and A in Rps802p and the 133aa: N in Rps801p and T in Rps802p. (**B**) Diagram illustrating the genetic construction of homogenic strains and chimeric strains. Details seen in Appendix A. (**C**) Quantification of the relative rps8 mRNA level normalized to act1 by qRT−PCR. (**D**) Quantification of the colony size reflecting growth rates among rps8 mutants. Cells derived from wild−type, rps8 paralog deletions, and chimeric strains, as well as homogenic strains, were planted at the density of one cell/cm^2^ by microscopic manipulation on YE + 5S plates, incubated at 29 °C for four days. Each colony is photo−documented on the 4th day, and colony diameters are measured with Photoshop software. At least 50 colonies are calculated for each strain. The representative colony for each strain is shown. Scale bar is 1 mm. (**E**) Polysome analysis by sucrose gradient−based centrifugation to assess the relative abundance of 40S, 60S, and 80S ribosomes and polysomes. (**F**) Comparison Rpl42/eL42 protein levels among the wild−type and rps8 genetic mutants by Western blotting. *p*−values were obtained by the two−tailed Fisher’s exact test. * *p* < 0.05; ** *p* < 0.01. (**G**) Comparison cell growth of the wild−type, rps8 paralog deletions, and chimeric strains, as well as homogenic strains in YE supplied with excess NH_4_Cl.

**Figure 4 cells-11-02381-f004:**
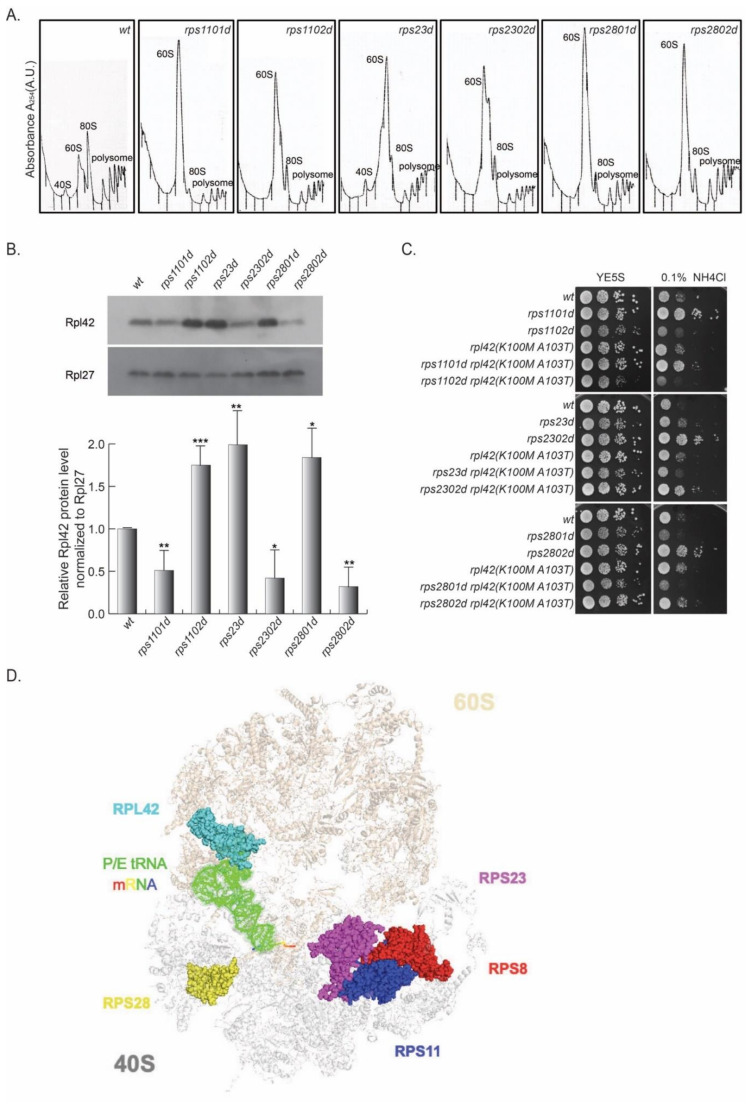
The differential Rpl42/eL42 protein level can also be detected in other 40S paralog deletions. (**A**,**B**) Comparison of the Rpl42/eL42 protein level between rps11, rps23, and rps28 paralog deletions by polysome analysis and Western blotting. Compared with wild−type cells, there were higher levels of Rpl42p/eL42 in rps1102d, rps23d, and rps2801d cells than in rps1101d, rps2302d, and rps2802d cells. *p*−values were obtained by the two−tailed Fisher’s exact test. * *p* < 0.05; ** *p* < 0.01; *** *p* < 0.005. (**C**) rpl42 also has specific genetic interactions with the rps11, rps23, and rps28 genes. By introducing the rpl42 (K100M A103T) mutation into the rps11, rps23, and rps28 paralog deletion strains, the phenotype of enhanced growth in the presence of excess NH_4_Cl of rps1101d, rps2302d, and rps2802d was reversed, similar to what was observed in rps802d cells. (**D**) Top view of the Saccharomyces pombe 80S ribosome in a complex with P/E site tRNA (green) and mRNA (rainbow). Rps8/eS8 (red), Rps11/uS17 (dark blue), Rps23/uS12 (purple), Rps28/eS28 (yellow), and Rpl42/eL42 (light blue) are shown in a transparent space-filling mode, while the large (60S) and small (40S) subunits are shown in brown shading and blue shading, respectively.

**Figure 5 cells-11-02381-f005:**
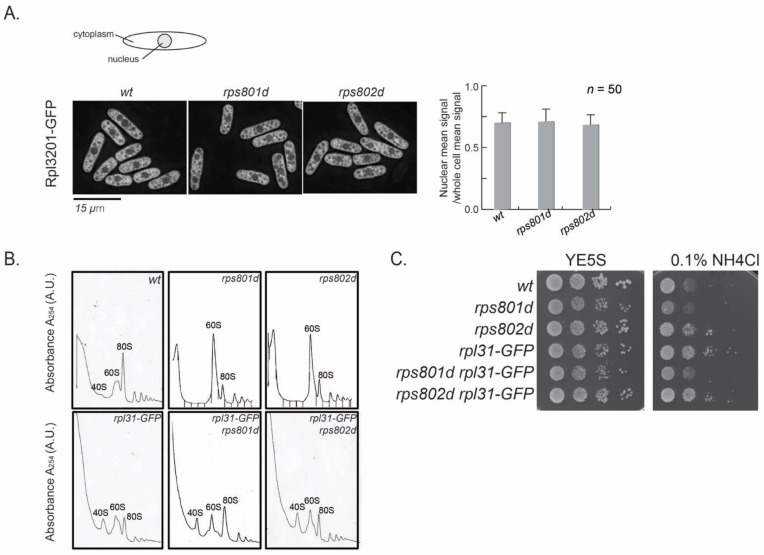
The 60S subunits accumulation is independent of the varied Rpl42/eL42 levels in *rps8* paralogs. (**A**) Rpl3201/eL32 was C−terminally tagged with GFP, and its localization and quantification of the GFP intensity in the nuclei were examined in the *wt* and *rps801d*, *rps802d*. Quantification of the GFP intensity density in the nucleus compared to the whole cell is shown for the experiment presented on the left. Quantification was performed using ImageJ on the maximum projections of the z−sectioning images. A representative image is shown in each case. Error bars represent the SD. (**B**) 60S subunits accumulation is alleviated by the GFP fusion of Rpl31/eL31 in *rps8* paralog deletion pairs by the polysome analysis. (**C**) Compromised 60S subunits accumulation is independent of other phenotypes seen in *rps8* paralog deletion pairs. The GFP fusion of Rpl31/eL31 did not affect the growth of the *rps8* paralog deletion pairs in YE supplied with NH_4_Cl.

## Data Availability

Strains and plasmids are available upon request. The authors affirm that all data necessary for confirming the conclusions of the article are present within the article, figures, tables, and Appendix A.

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
