# Peer review of "Differential Paralog-Specific Expression of Multiple Small Subunit Proteins Cause Variations in Rpl42/eL42 Incorporation in Ribosome in Fission Yeast"

_cells, 2022, doi:10.3390/cells11152381_

Round 1
Reviewer 1 Report
This manuscript by Li and colleagues describes a very nice study about some roads leading to ribosome diversity in yeast. By modifying the expression of different Rp gene paralogs, they show the downstream impact on the expression of other rps, and on the cellular phenotype. Most of these genetic modifications generate an imbalance in the production of the two ribosomal subunits, thus altering the availability of 80S ribosomes for translation.
My overall feeling is that the manuscript offers very good perspectives, which, however, are, in many occasions, overstated. Indeed, extra experiments should be performed to support the claims of the authors. Actually, no direct proof of what they claim- that, the deletions of different rps paralog genes cause different levels of Rpl42p/eL42 incorporation into actively translating ribosomes, and that this indicates that RPS expression levels may affect Rpl42p/eL42 incorporation in mature ribosomes.
Following are my specific comments, in order of reference to the text (not of importance):
- Abstract- We 16 present extensive genetic evidence in the fission yeast S.pombe, that independent, paralogous genes 17 with subtle difference in sequence but encoding the same ribosome protein component, many are 18 functionally different. This sentence has something wrong.
- In table S1 the legend is missing: how to interpret correctly NA, bi-stable, etc?
- Fig. 1C Because this technique is only qualitative, at a maximum semi-quantitative, it would be useful to compare the 40S, 60S and 80S areas under the peaks to total area. Actually, in the mutant profiles the 40S peak seems to be completely lacking, and some control for the correct fractionation should be added (e.g. by extracting proteins from the polysomal fractions and blotting for at least a couple of RPSs, to check for their correct distribution in the different fractions as compared to the profile). The same goes with the profiles in fig 3E/F. Protein abundance should be shown also for pre-polysomal fractions.
- Fig. 2B what is in each lane of the WB?
- Fig S3D is missing. In S3C the WB bands quantification is also missing. What are the labels on the lanes? (OP stands for…?). OE may not be visible in whole cell extracts due to the fact that most of the protein is anyway embedded in ribosomes (which have a half-life of several days). How about looking at the “free” protein after separation of ribosomes and ribosomal subunits?
- Lines 381-83: Together, these results showed that ribosome small 381 subunit protein Rps8/eS8 paralogs have a strong genetic interaction with the ribosome 382 large subunit protein Rpl42/eL42 in response to amino acid starvation.
Is this possible interaction only genetic? How are these proteins positioned within the 80S ribosomes? Do they have ay physical or functional interaction? These points should be considered, in my opinion.
- Fig. 5a: what are we looking at? Nuclei or whole organisms? This is not clearly indicated in the caption. IN the second panel there is actually more fluorescence. How is this explained? Also, the claim would be better supported by a nuclear/cytoplasmic fractionation experiment followed by WB on endogenous protein. The best option would be to load the cytoplasms on sucrose gradients for polysome fractionation, and separately lyse the nuclei. Then load everything on the same WB to check for protein distribution and abundance.
- Figs. 4/5. How are rpl3201 and rpl31 picked for these experiments? This is not clearly explained and it is quite difficult to follow the narrative of this manuscript, whit this kind of logical gaps.
- Discussion: these results imply that, Rpl42p/eL42 in actively translating ribosomes pu-rified from wild type cells are at the sub-stoichiometry level. By extension, ribosomes in 573 wild type cells may exist in two forms – those with and those without Rpl42p/eL42, which 574 probably are at a balanced ratio. Accurate measurement of the sub-stoichiometric levels 575 of Rpl42p/eL42 in ribosomes need to be carried out in the future by more quantitative 576 approaches such as quantitative proteomic mass spectrometry or cryo-EM in pombe cells 577 at various genetic settings or physiological conditions These statements can only be hypothesized based on the data shown herein, and direct MS on purified ribosomes needs to be performed to support this claim. Based on this, a more cautious phrasing is preferable.
- The authors hypothesize that rpl42 abundance in ribosomes may affect translation efficiency. However, translation efficiency was not tested (apart from polysome profiles, which only give a suggestion about translation, but may as well be stalling and produce no protein). Also, phrasing should be changed. (lines 588-90)
- Fig. S7 and S9 show results which are only brought out in the discussion. This is weird, results should be first shown in the results section, then discussed in discussion section.
- Figure s8 in my opinion should be included among the main figures. It should be implemented, to show the catalytically relevant portions of the ribosome (e.g. A,P,E sites, peptide exit tunnel, mRNA tunnel, ecc)
- Lines 613-617: how about different ribosomes translating with different efficiencies the mRNAs encoding for specific ribosomal proteins? You could even check for this, if you extracted and quantified the transcripts in your polysomal fractions and in total rnas…
Reviewer 2 Report
The authors address the whether or not paralog-specific phenotypes in yeast are due to the heterogeneity in the protein composition of ribosomes or due to the differences in ribosome concentration, or both. The authors used a genetic approach used to generate the mutants and microbiological approaches were used to characterize the functions of the phenotypes. Ribosome composition was analyzed by qRT-PCR, sucrose gradient ultracentrigation and mass spectrometry. They found that deletions of the Rp8/eS8 paralog pair lead to mutant cells with that differences in ribosome composition with regard to the levels of Rp142p/eL42 whereas protein sequence variation was not the cause of the observed differences. The authors provide strong experimental evidence to back up this finding.This key finidng merits publication because it is of broad interest and high biological signficance.
The paper is well written. Nonetheless, one improvement would be to include an introductory paragraph that maps out the six sections that flow so that the reader is has an inkling of what is to come.
Author Response
We appreciate the Reviewer 2's positive comment. We do have an introductory paragraph in the the end of the Introduction Part (Page 2-3, Line 109-121) to sum up our main findings.
Round 2
Reviewer 1 Report
I thank the authors for considering most of my suggestions.
I still have a few minor observations :
Figure S4C. What does “purified ribosomes” mean? I did not find any reference to the purification method in materials and methods section. If the authors refer to polysomal fractions, they should state it clearly in the figure legend.
In Fig. 2B the annotations in the WB are still missing
Fig. 5a has not been replaced.
Author Response
We thank for the Reviewer's specific observations. Here are our responses:
- We have corrected the "purified ribosome" as the "polysome fractions" in Figure S4C. And stated it clearly in the figure legend in supplemental files.
- We apologized for the missing annotation of WB in Fig 2B. And we have added the annotation and made clear statement in figure legend (Page 9, Line 365).
- We have replaced the images in Fig5a to avoid misunderstanding.